# OpenReview forum: "Rethinking Benign Relearning: Syntax as the Hidden Driver of Unlearning Failures"
_ICLR.cc/2026/Conference — ICLR 2026 Poster_

### Official Review · Reviewer_Ez5r · 2025-10-28

**Soundness:** 3
**Presentation:** 3
**Contribution:** 3
**Rating:** 6
**Confidence:** 3

**Summary:**

This paper studies why unlearning comes back after benign fine-tuning (“benign relearning”) in LLMs. Contrary to the common belief that topic overlap (e.g., Harry-Potter-related text) is the main culprit, the authors argue and empirically show that syntactic similarity (shared surface forms/templates) is the real driver: fine-tuning on sentences with the same structure (even about different entities) reactivates the forgotten content. They formalize and test this on BLUR/TOFU-style setups, diagnose evaluation confounds in prior relearning studies, and propose a simple mitigation, syntactic diversification (paraphrasing forget queries into varied structures before unlearning). This diversification both reduces relearning, speeds up forgetting, and improves utility trade-offs.

**Strengths:**

1. The paper shows syntactic similarity (in the query), not topicality, is the consistent relearning driver across methods (GA, NPO, SCRUB) and datasets. The paper also identifies evaluation confounds (dataset size -> step budget i.e., non-monotonic training trajectories) that can make topicality look stronger than it is, then re-evaluates with a step-standardized protocol. This corrects the narrative and is a valuable insight for the community.

2. The analysis provided  with the Heatmaps, relearning vs unlearning steps, demonstrating the and analyses show syntactically similar relearn sets recover forgotten names more strongly than topically relevant ones (measured via keyword presence / ROUGE-L to base). Representation- and gradient-similarity analyses support the mechanism. Overall syntactic overlap aligns hidden states and gradients with the target set after unlearning, explaining recovery.

3. Syntactic diversification (multi-paraphrase forget queries via GPT-4o) reduces template rigidity, balances loss between templates and keywords, delays/attenuates relearning, requires fewer unlearning steps, and improves utility (e.g., Real Authors, World Facts and Retain split). That’s a rare win-win (better robustness with less utility damage).

**Weaknesses:**

1. The approach relies on GPT-4o paraphrasing. What is the cost/latency at unlearning time for large forget sets, and does quality vary by domain or language? A scaling/cost analysis and a cheaper in-house paraphrase baseline would help adoption.

2. Keyword-based relearn success rate captures name reappearance but may miss partial leakage or paraphrastic leakage. Similarly, ROUGE-L to base captures surface similarity but not factual equivalence. Including embedding-based and judge-LM evaluations would strengthen claims.

3. The overall message in the paper (templates get suppressed more than keywords during unlearning) is compelling, but stronger causal tests (e.g., controlled template injection/removal, counterfactual templates, layerwise intervention) would be needed to further substantiate this analysis.

**Questions:**

1. Can you report results with richer syntactic similarity measures (e.g., tree edit distance, template mining) and show which correlates best with relearning?

2. Can you add judge-LM / embedding-based leakage metrics to complement keyword/ROUGE and analyze disagreements, if not then can you help explain why such an analysis could not be performed?

---

> ### Author Response · Authors · 2025-11-19
> **Rebuttal - Part I**
>
> We thank Reviewer Ez5r for the thorough and constructive review. We appreciate the reviewer’s recognition of our analysis on syntactic similarity and the utility of syntactic diversification, and we address the reviewer’s suggestions regarding cost analysis, richer syntactic metrics, and stronger causal tests in the responses below.
>
> ---
>
> **1. Dependence on GPT-4o Paraphrasing and Scalability**
>
> To test whether syntactic diversification relies on large proprietary models, we repeated the full diversification pipeline using Llama-3-8B instead of GPT-4o. We observe no degradation: diversification with Llama-3-8B achieves nearly the same level of relearning resistance as GPT-4o across all unlearn–relearn configurations. Moreover, since diversification rewrites existing content into alternative syntactic forms rather than generating new knowledge, domain and language difficulty has limited impact on the transformation. The benefit arises from generating multiple syntactic variants of the same forget query while preserving semantics, a capability that even smaller open-source models provide reliably.
>
> Regarding scalability, syntactic diversification is a one-time preprocessing step whose cost grows linearly with the size of $D_{\text{forget}}$. Using Llama-3-8B substantially reduces diversification cost. Moreover, as shown in Figure 9, diversification reduces the number of gradient update steps needed for unlearning, thereby decreasing total unlearning time and computation.
>
> Together, these results show that syntactic diversification is practical, cost-efficient, and scalable, and does not need to rely exclusively on high-end commercial models.
>
> * Relearn success rate with and without diversification (lower = stronger resistance)
> | Relearn steps | No Diversification | GPT-4o Diversification | Llama-3-8B Diversification |
> |---------------|--------------------|-------------------------|----------------------------|
> | **0 steps**| 0.0                | 0.0                     | 0.0                        |
> | **11 steps** | 0.2                | 0.0                     | 0.0                        |
> | **23 steps** | 0.8                | 0.0                     | 0.0                        |
> | **35 steps** | 0.7                | 0.0                     | 0.0                        |
> | **47 steps** | 0.7                | 0.0                     | 0.0                        |
>
> ---
>
> **2. Additional semantic leakage metrics**
>
> We thank the reviewer for the suggestion and performed additional evaluations using two stronger semantic leakage detectors designed to capture paraphrastic or partial leakage that keyword matching or ROUGE-L may miss: embedding-based cosine similarity, and LLM-as-judge leakage evaluation. Across all three metrics, we observe extremely high consistency, with Pearson correlations above 0.99:
>
> * Pearson correlations
> | Correlation | Keyword | Cosine | LLM |
> |-------------|---------|--------|-----|
> | **Keyword**     | 1.000| 0.995  | 0.992 |
> | **Cosine-Sim**     | 0.995   | 1.000| 0.997 |
> | **LLM-Judge**         | 0.992   | 0.997  | 1.000|
>
> To complement these correlation results, we also report the actual leakage values measured by each detector. All three metrics unanimously show that the syntactically similar relearn set produces substantially higher leakage than the topically relevant set across all unlearning steps.
>
> * Semantic leakage across three detectors (higher = more leakage)
> | Relearn Set Type      | Relearn Steps | Keyword | Cosine Similarity | LLM-as-Judge |
> |-----------------------|---------------|---------|--------------------|--------------|
> | **Topically relevant**    | **11 steps**      | 0.0     | 0.0073             | 0.0          |
> |  &nbsp;   | **23 steps**      | 0.0     | 0.0150             | 0.0          |
> |    &nbsp;  | **35 steps**     | 0.0     | 0.0159             | 0.0          |
> |   &nbsp;   | **47 steps**      | 0.0     | 0.0251             | 0.0          |
> | **Syntactically similar** | **11 steps**      | 0.2     | 0.2488             | 0.3          |
> | &nbsp; | **23 steps**      | 0.8     | 0.7113             | 0.9          |
> | &nbsp; | **35 steps**      | 0.7     | 0.7072             | 1.0          |
> | &nbsp; | **47 steps**      | 0.7     | 0.7254             | 0.9          |
>
> These results show that our conclusions about relearning remain unchanged regardless of the metric used. Even under stronger semantic evaluations, all detectors yield the same ordering and overall patterns of leakage. Thus, the observed relearning behavior is metric-agnostic, and our findings hold under keyword-based, embedding-based, and judge-LM evaluations alike.

---

> ### Author Response · Authors · 2025-11-19
> **Rebuttal - Part II**
>
> **3. Causal test validating that templates are suppressed more strongly than keywords**
>
> To directly test whether unlearning suppresses templates more aggressively than keywords, we conducted a controlled template-injection experiment. The original evaluation prompt is:
> >[INST] What is the full name of the author born in Kuwait City, Kuwait on 08/09/1956? [/INST]
>
> To isolate the effect of the template, we constructed a counterfactual version in which we forcefully provide the answer-prefix template while omitting the actual keyword (the author’s name):
> >[INST] What is the full name of the author born in Kuwait City, Kuwait on 08/09/1956? [/INST]The full name of the fictitious author born in Kuwait City, Kuwait on the 8th of September, 1956 is
>
> This prefix removes the need for the model to produce the template structure itself. What remains is whether the model can still recall the target keyword that should have been forgotten.
>
> Across all unlearning steps, the model continues to output the correct keyword almost perfectly, despite strong suppression of free-form generation.
>
> * Attack success rate under template injection (higher = more leakage)
> | Unlearning Steps | Base Query | Template-Injection |
> |------------------|------------|--------------------|
> | **25 steps**| 0.6        | 0.9                |
> | **31 steps**| 0.7        | 0.9                |
> | **37 steps**| 0.6        | 0.9                |
> | **43 steps**| 0.5        | 0.9                |
>
> This controlled intervention provides direct causal evidence that:
> 1. unlearning primarily suppresses *template structures*, but
> 2. keyword-level factual information remains largely intact, even after extensive unlearning.
>
> **Effect of our method (syntactic diversification) under the same causal test**
>
> To verify that our method explicitly reduces keyword-level leakage, we repeated the same template-injection experiment using the diversified forget set $D’_{\text{forget}}$. In this setting, both template-based and keyword-based cues are substantially suppressed, and leakage drops close to zero across all unlearning steps.
>
> * Attack success rate under template injection with diversification (lower = stronger forgetting)
> | Unlearning Steps | Base Query | Template-Injection |
> |------------------|------------|--------------------|
> |**25 steps**| 0.3        | 0.3                |
> |**31 steps**| 0.0        | 0.0                |
> |**37 steps**| 0.0        | 0.0                |
> |**43 steps**| 0.0        | 0.0                |
>
> These results show that syntactic diversification removes the stable surface patterns that enable benign relearning, thereby suppressing both template and keyword. This further supports our core claim: while standard unlearning predominantly suppresses templates, diversification ensures that both template structure and factual information are effectively removed.

---

> ### Author Response · Authors · 2025-11-19
> **Rebuttal - Part III**
>
> **4. Results with richer syntactic similarity measures**
>
> To ensure that our findings do not depend on normalized Levenshtein distance, we conducted additional analyses using two complementary syntactic similarity metrics: template-mining similarity [1], and parse-tree similarity [2].
> Across all metrics, $D_{\text{relearn}}^{\text{syntactic}}$ consistently shows substantially higher similarity to $D_{\text{target}}$ than $D_{\text{relearn}}^{\text{topic}}$, confirming that the structural distinction between the two relearn sets is robust and not tied to any specific metric.
>
> * Syntactic similarities measured by three metrics
> | Relearn Set Type | Template Mining | Parse Tree | Levenshtein |
> |------------------|-----------------|------------|-------------|
> | $D_{\text{relearn}}^{\text{topic}}$          | 0.3376          | 0.1541     | 0.2349      |
> | $D_{\text{relearn}}^{\text{syntactic}}$        | 0.6365          | 0.5040     | 0.4513      |
>
> To further verify metric-independence, we constructed three Top-190 relearn sets by selecting samples most similar to $D_{\text{target}}$ under each metric and evaluated their relearning strength. All three metric-specific relearn sets exhibit nearly identical patterns of strong relearning.
>
> * Relearn success rate of metric-specific relearn sets
> | Unlearning Steps | Template-based | Parse-tree-based | Levenshtein-based |
> | ---------------- | -------------- | ---------------- | ----------------- |
> |**31 steps**| 0.8            | 0.8              | 0.8               |
> |**37 steps**| 0.7            | 0.7              | 0.7               |
> |**43 steps**| 0.7            | 0.7              | 0.7               |
> |**50 steps**| 0.6            | 0.7              | 0.6               |
>
> These results show that relearning consistently emerges whenever the relearn set is syntactically close to $D_{\text{target}}$, regardless of whether similarity is measured via templates, parse-tree structure, or Levenshtein distance. This demonstrates that benign relearning is metric-agnostic and that syntactic proximity to $D_{\text{target}}$ is the key mechanism driving the effect.
>
> [1] Ding, Y., Chowdhury, G., & Foo, S. (1999, November). Template mining for the extraction of citation from digital documents. In Proceedings of the Second Asian Digital Library Conference, Taiwan (pp. 47-62).\
> [2] Collins, M., & Duffy, N. (2001). Convolution kernels for natural language. Advances in neural information processing systems, 14.

---

> > ### Comment · Reviewer_Ez5r · 2025-11-24
> > **Thanks for the response!!**
> >
> > Thank you for the detailed and thoughtful responses. I appreciate the additional analyses, especially the ablation evaluation with Llama-3-8B, the inclusion of embedding-based and LLM-judge leakage metrics, the controlled template-injection experiment, and the metric-agnostic syntactic similarity results. These additions substantially strengthen the paper and directly address the concerns raised in my review.
> >
> > I encourage the authors to incorporate these analyses (including key figures/tables or summaries) into the main paper or appendix so the broader community can benefit from the clarity they provide. I would also request that the authors clarify precisely what is being reported and how each quantity is computed for the numbers shown in the tables. For example, it would be helpful to elaborate on “Attack success rate under template injection (higher = more leakage)” and other table headings.
> >
> > I will update my score once the discussion concludes and after considering all reviewers’ comments and responses.

---

> > > ### Author Response · Authors · 2025-11-26
> > > **Response to Reviewer Ez5r**
> > >
> > > Thank you for your thoughtful follow-up and for recognizing the value of the additional analyses. We have incorporated all relevant figures, tables, and clarifications into the appendix, explicitly detailing what each metric measures and how all reported quantities are computed.
> > >
> > > ---
> > >
> > > **1. Dependence on GPT-4o Paraphrasing and Scalability**
> > >
> > > Our analysis on dependence on GPT-4o paraphrasing and scalability is included in **Appendix G.4**, where we replicate the syntactic diversification pipeline using the smaller open-source Llama-3-8B and show that it achieves the same level of relearning resistance as GPT-4o while significantly reducing preprocessing cost.
> > >
> > > ---
> > >
> > > **2. Additional semantic leakage metrics**
> > >
> > > We evaluate embedding-based and LLM-based evaluation metrics using the following procedures.
> > > * **Cosine similarity**: Following [1], we extract Sentence-BERT [2] embeddings of the model’s outputs before and after unlearning and compute their cosine similarity. Higher similarity indicates stronger semantic retention of the forgotten content.
> > >
> > > * **LLM-as-judge**: We compare the unlearned model’s output against the model's output before unlearning using GPT-4o. The judge assigns a score between 0 and 1, where higher values indicate a closer reproduction of the forgotten answer.
> > >
> > > The additional evaluation metrics and their correlations are provided in **Appendix H**.
> > >
> > > [1] Xiaojian Yuan, Tianyu Pang, Chao Du, Kejiang Chen, Weiming Zhang, and Min Lin. A closer look at machine unlearning for large language models. In ICLR, 2025. \
> > > [2] Nils Reimers and Iryna Gurevych. Sentence-bert: Sentence embeddings using siamese bert- networks. In EMNLP, 2019.
> > >
> > > ---
> > >
> > > **3. Causal test validating that templates are suppressed more strongly than keywords**
> > >
> > > In the template injection experiment, Attack Success Rate (ASR) uses the same keyword-based scoring rule as the Relearn Success Rate, but is computed on the unlearned model to quantify leakage. For each query, we assign a score of 1 if the model’s output contains the exact target keyword and 0 otherwise. A higher ASR indicates that the unlearned model is still able to regenerate the forgotten keyword, reflecting a greater failure of unlearning.
> > >
> > > The causal test showing that unlearning suppresses templates more strongly than keywords is presented in **Appendix F**, which also includes the definition of ASR used in our evaluation.
> > >
> > > ---
> > >
> > > **4. Results with richer syntactic similarity measures**
> > >
> > > We evaluate additional syntactic similarity metrics, Template mining and Parse tree, using the following procedures.
> > >
> > > * **Template mining**: We compute template mining similarity using POS-based templates. Each sentence is converted into a POS-tag sequence, where each tag represents the grammatical role of a word. Similarity is then measured by comparing the overlap between the POS tokens of two sentences, and a higher score indicates that the sentences share more of the same surface-level syntactic patterns.
> > >
> > > * **Parse tree**: We compare syntactic structure using POS-based parse trees by counting how many subtree fragments two trees share. The similarity score increases when the trees contain more common subtrees, reflecting a closer match in their syntactic structure.
> > >
> > > The descriptions of these syntactic similarity metrics and results are added in **Appendix I**.

---

### Official Review · Reviewer_anpb · 2025-10-30

**Soundness:** 3
**Presentation:** 4
**Contribution:** 3
**Rating:** 6
**Confidence:** 4

**Summary:**

This paper delves into a key and perplexing phenomenon in the field of LLM unlearning—benign relearning.
The core argument of this paper is that the primary driver of benign relearning is not, as previously thought, topical relevance, but rather syntactic similarity. The authors first use rigorous experimental design to identify confounding variables in the evaluation methods of previous benchmarks and then correct these experiments, finding that the role of topic relevance is overestimated.

To verify their core hypothesis, the authors construct two relearning datasets: "topically related but syntactically different" and "topically unrelated but syntactically similar." Experimental results show that syntactically similar data triggers information retrieval more effectively. The paper further analyzes the underlying mechanism from the perspectives of representation and gradients, revealing that syntactically similar data and the forgetting target are highly aligned within the model, and that the standard forgetting process suppresses "templates" rather than "keywords," thus leaving structural "backdoors."

Based on this finding, the paper proposes a simple yet effective solution—syntactic diversification. This method enriches the syntactic structure of the forgetting set through paraphrasing before forgetting. Experiments demonstrate that this method not only effectively inhibits benign relearning but also accelerates the forgetting process and significantly reduces the impairment to the model's generalizability.

**Strengths:**

1. The problem is clearly and significantly addressed. The authors challenge mainstream understanding in the field (topic relevance-driven) and propose a novel and insightful perspective (syntactic similarity-driven), crucial for understanding the failure of forgetting mechanisms.

2. The experimental design is rigorous and comprehensive, ensuring fair and rigorous evaluation. Evaluation confounding in BLUR is identified and eliminated, the number of steps is standardized, and the optimal result is chosen, making the conclusions more reliable.

3. The defense methods are simple and practical. Syntactic diversification requires no modification to the optimizer or model structure, yielding significant results (slower and weaker relapses, fewer forgetting steps, and better utility), and demonstrating robustness.

4. The writing is clear and logically coherent: the paper's structure is clear and easy for readers to understand.

**Weaknesses:**

1. Limitations of the syntactic similarity metric. The paper uses normalized Levenshtein distance as a measure of syntactic similarity. While this is a simple and effective character-level metric, it may fail to capture more abstract and deeper syntactic structures (such as the structural similarity of parse trees). It is suggested to explore using more sophisticated syntactic analysis tools to measure syntactic similarity, which could reveal more subtle mechanisms.

2. Cost of the proposed solution. The proposed solution relies on the robust GPT-4o model to generate syntactic variants. It is suggested to conduct a short ablation experiment to explore the effects of using smaller, more cost-effective open-source models (such as Llama-3-8B itself) or simpler methods for syntactic diversification.

**Questions:**

see weakness

---

> ### Author Response · Authors · 2025-11-19
> **Rebuttal - Part I**
>
> We thank Reviewer anpb for the positive and insightful evaluation. We appreciate the reviewer’s recognition of our problem formulation, experimental rigor, and the practicality of our proposed method. Below, we address the reviewer’s suggestions regarding syntactic similarity metrics and the cost analysis of syntactic diversification.
>
> ---
>
> **1. Limitations of the syntactic similarity metric**
>
> We appreciate the reviewer’s suggestion regarding the limitations of relying solely on Levenshtein distance. To ensure that our conclusions do not depend on any single similarity measure, we conducted additional analyses using two complementary syntactic indicators: template mining similarity [1] and parse tree similarity [2]. These metrics capture phrase-level and structural properties that Levenshtein distance does not model, providing a broader characterization of syntactic relatedness.
>
> Across all three metrics, $D_{\text{relearn}}^{\text{syntactic}}$ consistently shows higher similarity to $D_{\text{target}}$ than $D_{\text{relearn}}^{\text{topic}}$, confirming that the distinction between the two relearn sets is robust and not an artifact of a particular metric.
>
> * Syntactic similarities measured by three metrics
> | Relearn Set Type | Template Mining | Parse Tree | Levenshtein |
> |------------------|-----------------|------------|-------------|
> | $D_{\text{relearn}}^{\text{topic}}$          | 0.3376          | 0.1541     | 0.2349      |
> | $D_{\text{relearn}}^{\text{syntactic}}$        | 0.6365          | 0.5040     | 0.4513      |
>
> To further verify that our conclusions do not depend on the choice of metric, we constructed three new relearn sets for the TOFU dataset by selecting the top 190 samples most similar to the target set under each metric. All three metric-specific relearn sets exhibit nearly identical patterns of strong relearning, confirming that the effect is robust and metric-agnostic.
>
> * Relearn success rate of metric-specific relearn sets (higher = more leakage)
> | Unlearning Steps | Template-based | Parse-tree-based | Levenshtein-based |
> | ---------------- | -------------- | ---------------- | ----------------- |
> |**31 steps**| 0.8            | 0.8              | 0.8               |
> | **37 steps**| 0.7            | 0.7              | 0.7               |
> |**43 steps** | 0.7            | 0.7              | 0.7               |
> | **50 steps**| 0.6            | 0.7              | 0.6               |
>
> Together, these results demonstrate that our conclusions are robust across syntactic metrics of varying granularity. Whether similarity is measured through templates, parse-tree structure, or edit-distance-based methods, relearning consistently emerges whenever the relearn set is syntactically close to $D_{\text{target}}$. This supports our central claim that syntactic proximity to $D_{\text{target}}$ is the fundamental mechanism underlying benign relearning.
>
> [1] Ding, Y., Chowdhury, G., & Foo, S. (1999, November). Template mining for the extraction of citation from digital documents. In Proceedings of the Second Asian Digital Library Conference, Taiwan (pp. 47-62).\
> [2] Collins, M., & Duffy, N. (2001). Convolution kernels for natural language. Advances in neural information processing systems, 14.

---

> ### Author Response · Authors · 2025-11-19
> **Rebuttal - Part II**
>
> **2. Cost of syntactic diversification**
>
> We appreciate the reviewer’s suggestion regarding the cost of syntactic diversification and conducted the proposed ablation. Specifically, we replaced GPT-4o with Llama-3-8B and applied the same diversification procedure using only this smaller open-source model. The results show that diversification with Llama-3-8B provides nearly the same level of relearning resistance as GPT-4o across all unlearn–relearn configurations. This demonstrates that our method does not rely on the expressive power of a large proprietary model; its effectiveness comes from generating multiple syntactic variants while preserving the underlying semantics, a capability that even smaller open-source models can supply.
>
> Overall, these findings indicate that syntactic diversification is practical, cost-efficient, and does not require access to high-end commercial models. Compared to unlearning without diversification, both Llama-3-8B and GPT-4o diversification substantially suppress relearning across all configurations.
>
> * Relearn success rate with and without diversification (lower = stronger resistance)
> | Relearn steps | No Diversification | GPT-4o Diversification | Llama-3-8B Diversification |
> |---------------|--------------------|-------------------------|----------------------------|
> | **0 steps**| 0.0                | 0.0                     | 0.0                        |
> | **11 steps**| 0.2                | 0.0                     | 0.0                        |
> | **23 steps**| 0.8                | 0.0                     | 0.0                        |
> | **35 steps**| 0.7                | 0.0                     | 0.0                        |
> | **47 steps** | 0.7                | 0.0                     | 0.0                        |

---

### Official Review · Reviewer_UHrD · 2025-11-02

**Soundness:** 2
**Presentation:** 4
**Contribution:** 3
**Rating:** 4
**Confidence:** 5

**Summary:**

This work investigated why benign relearning happens when using gradient based heuristic for unlearning LLM. Different from prior explanations, this work observed that syntactic similarity is the primary driver of why relearning works instead of topical relevance. The paper analyzes the BLUR dataset and showed that the relearn success rate is mainly due to the syntactic similarity, evaluated by the cosine similarity between relearn set and target set. The paper further proposes a new way of unlearning by using GPT to rewrite the forget set so that the forget set is syntactically different from the target set, which effectively limits the power of relearning.

**Strengths:**

- The paper is well written.
- The paper provides a new explanation of the success of relearning in the context of LLM unlearning, which is important for understanding in this community.

**Weaknesses:**

- One thing that has been significantly underlooked in this paper is **between which two sets** should syntactic similarity be looked at. There are three sets: unlearning set $D_{forget}$, eval set $D_{target}$, relearn set $D_{relearn}$. Section 5 characterizes the syntactic similarity between $D_{target}$ and $D_{relearn}$, but there are other dimensions: syntactic similarity between $D_{forget}$ and $D_{relearn}$ and syntactic similarity between $D_{forget}$ and $D_{target}$. In the TOFU case, an implicit assumption is $D_{target}$ and $D_{forget}$ high overlaps. Therefore, $D_{relearn}^{syntactic}$ is syntactically different enough from both $D_{target}$ and $D_{forget}$. However, such assumption might not be true for e.g. WMDP, where $D_{forget}$ is pub-med articles and $D_{target}$ are some expert drafted questions, not necessarily about the verbatims in the articles themselves. From the current analysis, what is missing is, **whether the success of relearning is due to syntactic similarity between $D_{target}$ and $D_{relearn}$ or $D_{forget}$ and $D_{relearn}$, or more complex among all three sets**.
- It is also important to make clear separation between knowledge unlearn (such as wmdp, where $D_{target}$ and $D_{forget}$ can usually be very syntactically different) and verbatim unlearn (such as tofu, where $D_{target}$ and $D_{forget}$ are potentially closer, but also try cases where you paraphrase $D_{target}$ itself) and investigate the above question under both cases.
- The robust unlearning part is less convincing to me. In practice, designing unlearn set given target set is unfair. The ideology should be a defender build an unlearn set with the purpose of defending against model outputting sensitive information, not against a set of fixed queries. Moreover, one can always rewrite the target set with GPT as well.
- Figure 8 is also less convincing. For TOFU, the original relearn set is a subset of the unlearn set. Now that if the forget set changes, shouldn't the relearn set also changes? Otherwise, this is not an apple-to-apple comparison as the adversarial in this case has less information.

**Questions:**

- Have the authors explore systematic analysis on knowledge unlearning task such as WMDP?
- I like this paper and think this paper provides good observations. The most important thing is a clearer and more systematic investigation to the weakness 1 above. That's the major issue I think the authors should address. The defense part in its current form is also not convincing as it is too rough and did not consider stronger adversarial. **I am willing to raise my score if both points have been further explored and explained in the rebuttal.**

---

> ### Author Response · Authors · 2025-11-19
> **Rebuttal - Part I**
>
> We thank Reviewer UHrD for the careful assessment and constructive questions. We appreciate the recognition of our contributions and address the reviewer’s concerns on syntactic similarity, dataset relationships, and the robust-unlearning setup in the responses below.
>
> ---
>
> **1. Clarifying the target set is not the eval set**
>
> We appreciate the reviewer’s question and would like to clarify the dataset definitions.
> In our formulation, the target set $D_{\text{target}}$ is not the evaluation set $D_{\text{eval}}$.
> Instead, $D_{\text{target}}$ is the subset of the forget set $D_{\text{forget}}$ whose information the unlearning procedure is intended to remove.
> A relearning process may or may not know the exact form of this target set; the key question we study is whether information in $D_{\text{target}}$ reappears after such relearning.
> In contrast, $D_{\text{eval}}$ is separate: it does not define what must be forgotten. It is used only as a diagnostic tool to test whether knowledge in $D_{\text{target}}$ resurfaces.
> This distinction matters for syntactic similarity: because relearning is evaluated based on whether $D_{\text{target}}$ resurfaces, the syntactic relationship most relevant to the phenomenon is between $D_{\text{relearn}}$  and $D_{\text{target}}$.
>
> ---
>
> **2. Which dataset pair actually drives relearning?**
>
> In our setup, relearning is determined by whether information in $D_{\text{target}}$ resurfaces, so the syntactic relationship that matters is between $D_{\text{relearn}}$ and $D_{\text{target}}$. TOFU can blur this distinction because its evaluation queries share the same template as $D_{\text{target}}$. To isolate the effect, we paraphrased all $D_{\text{eval}}$ so that they were syntactically different from both relearn sets.
> Even after 50 unlearning steps, $D_{\text{relearn}}^{\text{syntactic}}$ still fully restored the forgotten information (fully recovered within 11 relearning steps), while $D_{\text{relearn}}^{\text{topic}}$ did not (failed to recover at all even with many more relearning steps). This shows that relearning is not driven by similarity to the evaluation queries but by syntactic proximity between $D_{\text{relearn}}$and $D_{\text{target}}$.

---

> ### Author Response · Authors · 2025-11-19
> **Rebuttal - Part II**
>
> **3. Systematic analysis on knowledge unlearning tasks such as WMDP**
>
> In knowledge unlearning tasks such as WMDP, where the forget set and the eval set differ substantially in surface form, it becomes possible to separate all relevant pairwise similarities. We therefore compared two relearn sets: one syntactically similar to $D_{\text{target}}$, and another similar to $D_{\text{eval}}$. As shown in the table below, the target-similar relearn set consistently induces much stronger relearning across all steps. This demonstrates that syntactic proximity to $D_{\text{target}}$, not to $D_{\text{eval}}$ is the dominant factor driving benign relearning in both knowledge-unlearning and verbatim-unlearning settings.
>
> * Relearn success rate across relearn steps (higher = more leakage)
> | Relearn Set    | Relearn Step 8 | Relearn Step 12 | Relearn Step 16 | Relearn Step 20 | Relearn Step 24 |
> |:--------------:|:--------------:|:---------------:|:---------------:|:---------------:|:---------------:|
> | **Target-similar** | 0.24           | 0.42            | 0.58            | 0.64            | 0.74            |
> | **Eval-similar**   | 0.20           | 0.28            | 0.42            | 0.46            | 0.52            |
>
> ---
>
> **4. Robust-unlearning setup and justification of the Figure 8 design**
>
> Thank you for raising this point. We first clarify the roles of the three datasets.
> In our framework, $D_{\text{target}}$ is a subset of $D_{\text{forget}}$: it is the portion of the forget set whose information unlearning is meant to erase.
> The evaluation set $D_{\text{eval}}$ is separate from both and is used only as a probe to test whether information from $D_{\text{target}}$ resurfaces after relearning.
> Importantly, $D_{\text{eval}}$ plays no role in constructing $D_{\text{forget}}$ or its diversified version
> .
> This remains true even when we paraphrase the forget set; diversification is performed on $D_{\text{forget}}$ alone and is independent of how evaluation queries are worded.
>
> Regarding Figure 8: the relearn set is not a subset of $D_{\text{forget}}$.
> It is drawn independently from $D_{\text{retain}}$ and selected only because it is syntactically similar to $D_{\text{target}}$.
> We keep this relearn set fixed across conditions to isolate the effect of syntactic diversification.
> By modifying only $D_{\text{forget}}$ and holding $D_{\text{relearn}}$ constant, the experiment cleanly attributes any behavioral differences to diversification rather than to differences in the relearn data.
>
> Finally, even under the scenario the reviewer suggests, where a relearner might change $D_{\text{relearn}}$ depending on how $D_{\text{forget}}$ is rewritten, our method remains robust.
> Figure 9 shows that when the original forget set is syntactically homogeneous, unlearning disproportionately suppresses template tokens, leaving keyword information under-suppressed and easily reactivated by any syntactically similar relearn set.
> In contrast, syntactic diversification forces unlearning to suppress the keyword information directly, preventing relearning regardless of which syntactically varied relearn data is used.
> Thus, our approach remains effective even if an attacker rewrites evaluation queries, substitutes different syntactic forms, or attempts to infer the structure of $D_{\text{forget}}$.

---

### Author Response · Authors · 2025-12-03
**Comment for AC by Authors**

We sincerely thank the reviewers and the ACs for their valuable time and thoughtful feedback. We would like to offer a brief update following our rebuttal and revised manuscript, and to make clear that we have carefully addressed all reviewer concerns through detailed clarifications, additional analyses, and new experiments.

---

**1. Clarifying Dataset Roles and Identifying the True Driver of Relearning (Reviewer UHrD)**

We clarified that the target set $D_{\text{target}}$ is the subset of the forget set $D_{\text{forget}}$ whose information the unlearning procedure aims to remove, while the evaluation set $D_{\text{eval}}$ is entirely separate and serves solely as a probe to test whether forgotten knowledge resurfaces. This distinction is crucial because relearning is determined entirely by whether knowledge in $D_{\text{target}}$ reappears.

To verify this, we showed through both fully paraphrased evaluation queries and the WMDP experiments that relearning arises only when $D_{\text{relearn}}$ is syntactically close to $D_{\text{target}}$, not to $D_{\text{eval}}$. These results confirm that proximity to $D_{\text{target}}$ is what fundamentally governs relearning.

**2. Addressing the limitations of single-metric syntactic similarity (Reviewer anpb, Reviewer Ez5r)**

Using template-mining and parse-tree similarities, we confirmed that $D_{\text{relearn}}^{\text{syntactic}}$ remains substantially closer to $D_{\text{target}}$, aligning with our Levenshtein-based metric. The metric-specific top-similar relearn sets then exhibited identical relearning behaviors across all metrics, demonstrating that syntactic proximity drives relearning independent of the similarity measure used.

**3. Demonstrating the practicality and efficiency of diversification (Reviewer anpb, Reviewer Ez5r)**

We reproduced the full diversification pipeline with Llama-3-8B and confirmed that diversification remains equally effective without relying on proprietary large models, demonstrating that it is both cost-efficient and substantially more resistant to relearning than unlearning without diversification.

**4. Validating Metric-Agnostic Leakage Behavior (Reviewer Ez5r)**

We evaluated leakage using cosine similarity and LLM-as-judge detectors, both showing over 0.99 correlation with our keyword-based metric. Across all detectors, we observed the same ordering and patterns of leakage, with syntactically similar relearn sets consistently producing higher leakage. These results show that relearning behavior is stable across various metrics and validate both the reliability of our detector and the central role of syntactic similarity.

**5. Causal test validating template vs. keyword suppression (Reviewer Ez5r)**

We provided direct causal evidence that unlearning suppresses template structures more strongly than keyword-level knowledge through the template-injection experiments. In these experiments, we supplied the answer-prefix template while withholding the target keyword to test whether the unlearned model could still recover the forgotten content.

Under standard unlearning with a structured forget set, the model readily recalls the keyword once the template structure is provided, but this recovery no longer occurs when the syntactic structure of the forget set is diversified. These results indicate that standard unlearning suppresses only the template, whereas diversification removes the keyword-level cues themselves and therefore prevents such recovery.

---

### Meta-Review · Area_Chair_CGBK · 2025-12-31

**Summary:**

This paper investigates the benign relearning problem in LLM unlearning. Contrary to the prevailing belief that topical overlap drives relearning, the authors provide strong empirical and mechanistic evidence that syntactic similarity is the primary driver. Through careful experimental redesign, the paper disentangles confounding factors in prior benchmarks and shows that syntactically similar data consistently reactivates forgotten knowledge even without topical overlap.

Building on this insight, the authors propose syntactic diversification, a simple and practical strategy that paraphrases forget queries into heterogeneous syntactic forms prior to unlearning. This approach significantly suppresses benign relearning, accelerates forgetting, and improves the trade-off between unlearning effectiveness and model utility.

Across reviewers, the paper is regarded as well written, technically sound, and conceptually insightful, offering a meaningful reframing of how unlearning failures should be understood. Initial concerns focused on dataset relationships, robustness of the syntactic similarity metric, fairness of the robust-unlearning setup, cost and scalability of diversification, and the strength of causal evidence. The authors provided thorough, technically convincing rebuttals, adding extensive new analyses and experiments that directly addressed these concerns. **The reviewers did not respond to the authors' rebuttal.**

**Reviewer Concerns:**

Concerns that have been addressed satisfactorily:
- In response to the first significant concern about which dataset similarities drive relearning raised by Reviewer UHrD: the authors clarified the distinction between the target set and evaluation set, and demonstrated through paraphrased evaluation queries and WMDP experiments that relearning is governed by syntactic proximity between the relearn set and the target set, rather than similarity to evaluation queries.
- In response to the second significant concern about the robustness and fairness of the robust-unlearning setup raised by Reviewer UHrD: the authors clarified the experimental design and showed that syntactic diversification remains effective even under stronger adversarial assumptions, including when attackers may rewrite queries.
- In response to concerns about reliance on a single syntactic similarity metric raised by Reviewers anpb and Ez5r: the authors added analyses using template-mining and parse-tree similarity, and demonstrated that relearning behavior is consistent across metrics, confirming that the findings are metric-agnostic.
- In response to cost and scalability concerns raised by Reviewers anpb and Ez5r: the authors replicated syntactic diversification using the open-source Llama-3-8B model and showed comparable effectiveness to GPT-4o, establishing that the method is practical and cost-efficient.
- In response to concerns about leakage evaluation strength raised by Reviewer Ez5r: the authors added embedding-based and LLM-as-judge leakage metrics, showing extremely high agreement across detectors and strengthening the validity of the evaluation.
- In response to requests for stronger causal evidence raised by Reviewer Ez5r: the authors introduced controlled template-injection experiments that provide direct causal evidence that standard unlearning suppresses templates more strongly than keywords, and that syntactic diversification mitigates this failure mode.

**Reviewer Scores:**

- Reviewer UHrD: Initially 4; he/she explicitly indicated that he/she would increase the score given that the two major concerns were fully addressed. As the rebuttal, to my opinion, has satisfactorily addressed the two major concerns, the reviewer likely increases his/her score.
- Reviewer anpb: Initially 6; praised conceptual insight and rigor, with suggestions fully addressed.
- Reviewer Ez5r: Initially 6; acknowledged that added experiments and analyses substantially strengthened the paper and he would like to update the score (did not update possibly because the leakage incident).

---

### Decision · Program_Chairs · 2026-01-26

Accept (Poster)